# Preparation of Nanoparticle-Loaded Extracellular Vesicles Using Direct Flow Filtration

**DOI:** 10.3390/pharmaceutics15051551

**Published:** 2023-05-20

**Authors:** Shomit Mansur, Shahriar Habib, Mikayla Hawkins, Spenser R. Brown, Steven T. Weinman, Yuping Bao

**Affiliations:** Chemical & Biological Engineering, The University of Alabama, Tuscaloosa, AL 35487, USA; smansur@crimson.ua.edu (S.M.); shabib1@crimson.ua.edu (S.H.); mahawkins4@crimson.ua.edu (M.H.); sbrown104@crimson.ua.edu (S.R.B.)

**Keywords:** biomimetic nanocarriers, drug delivery, nanoparticle-loaded EVs, direct flow filtration, extracellular vesicle protein markers, cellular uptake

## Abstract

Extracellular vesicles (EVs) have shown great potential as cell-free therapeutics and biomimetic nanocarriers for drug delivery. However, the potential of EVs is limited by scalable, reproducible production and in vivo tracking after delivery. Here, we report the preparation of quercetin-iron complex nanoparticle-loaded EVs derived from a breast cancer cell line, MDA-MB-231br, using direct flow filtration. The morphology and size of the nanoparticle-loaded EVs were characterized using transmission electron microscopy and dynamic light scattering. The SDS-PAGE gel electrophoresis of those EVs showed several protein bands in the range of 20–100 kDa. The analysis of EV protein markers by a semi-quantitative antibody array confirmed the presence of several typical EV markers, such as ALIX, TSG101, CD63, and CD81. Our EV yield quantification suggested a significant yield increase in direct flow filtration compared with ultracentrifugation. Subsequently, we compared the cellular uptake behaviors of nanoparticle-loaded EVs with free nanoparticles using MDA-MB-231br cell line. Iron staining studies indicated that free nanoparticles were taken up by cells via endocytosis and localized at a certain area within the cells while uniform iron staining across cells was observed for cells treated with nanoparticle-loaded EVs. Our studies demonstrate the feasibility of using direct flow filtration for the production of nanoparticle-loaded EVs from cancer cells. The cellular uptake studies suggested the possibility of deeper penetration of the nanocarriers because the cancer cells readily took up the quercetin-iron complex nanoparticles, and then released nanoparticle-loaded EVs, which can be further delivered to regional cells.

## 1. Introduction

Extracellular vesicles (EVs) have attracted much attention as cell-free therapeutics [1,2,3] or drug delivery vehicles for various diseases [4,5,6]. They have demonstrated functions in regulating cell–cell communication [7,8], directing cell behaviors (e.g., cell proliferation [9], and cell migration [10,11], and crossing biological barriers [12]. Depending on the origin, there are three types of EVs with different sizes and molecular cargos [13,14,15,16], such as 50–150 nm diameter for smaller EVs, 100–1000 nm for microvesicles, and 500–2000 nm for apoptotic bodies [14,16]. In addition, their typical protein markers are also different [14,16,17], where EVs from endocytic cells express ALIX, TSG101, and tetraspanins (CD81, CD63, CD9); microvesicles contain flotillin, integrins, selectins, and CD40; and apoptotic bodies have Annexin V and phosphatidylserine. EVs have been studied for various applications [18,19,20]; however, the clinical translation of the potential benefits of EVs is limited by several practical challenges, such as in vivo tracking, and scalable and reproducible production [20,21].

First, it is challenging to effectively track EVs for therapeutic application after administration [22,23]. To address this issue, numerous studies have been performed to incorporate imaging probes to achieve in vivo tracking [23]. Therefore, loading EVs with imaging agents provides a great method for in vivo tracking. Second, the high heterogeneity of EVs makes reproducible preparation of EVs a major issue [20,21]. Clinical applications of EVs depends on the ability to isolate EV populations with specific sizes, biomolecular compositions, biological functions, and biodistribution. A failure to achieve this uniformity can result in compromised therapeutic effects and adverse outcomes [24]. 

Different techniques have been utilized for EV isolation based on their size and surface protein markers from cell culture media or biological fluids [25]. Ultracentrifugation (UC) is the most commonly used technique for EV isolation [26,27,28]. However, it suffers from reproducibility [29] and low yield [28,30]. Additionally, size exclusion chromatography, membrane filtration, and membrane affinity spin columns were explored to isolate EVs [31]. Among those techniques, membrane flow filtration has shown great potential in the preparation of EVs with high scalability, such as tangential flow filtration (TFF) [32,33,34,35]. During TFF, a feed solution containing EVs flows over a selective membrane with specific pore size, where smaller impurities are removed from the feed solution and the EVs are retained [36,37]. This technique has been applied to isolate EVs from large volumes of body fluids with high batch-to-batch reproducibility [32]. The side-by-side comparison with UC suggested a higher yield and better reproducibility of membrane filtration than UC [32]. Direct flow filtration (DFF) under constant stirring serves as an alternative to TFF [38,39]. During DFF, a feed solution containing EVs flows perpendicular to the membrane surface through the membrane pores where the applied pressure drives molecules smaller than the membrane pore size through the membrane and the targeted EVs (larger than the membrane pores) are captured on the membrane surface. TFF and DFF are similar in that they both have the same separation mechanism, sieving based on size. Compared with TFF, DFF filters the entire feed solution while TFF filters only ~50–70% of the feed solution [40]. Membrane filtration also allows for large-scale isolation of extracellular vesicles from cell culture with high purity [37]. For example, scalable production of biologically active EVs from mesenchymal stem cells was demonstrated using TFF with higher yield and improved activity [41]. However, flow filtration has not been explored for preparing nanoparticle-loaded EVs. In particular, DFF was less studied for EV isolation compared with TFF. 

In this paper, we report the preparation of quercetin-iron complex nanoparticle-loaded EVs using DFF. The quercetin-iron complex nanoparticles (QFeNPs) were synthesized by crosslinking Q-Fe complexes with excess iron ions. The morphology and size of the NPs were characterized by transmission electron microscopy (TEM) and dynamic light scattering (DLS). The ligand coordination was verified by Fourier transform infrared spectroscopy (FTIR). Subsequently, NP-loaded EVs were prepared by treating breast cancer cells, MDA-MB231br, with NPs under a stressed condition. Finally, the NP-loaded EVs were isolated using DFF. SDS-PAGE gel electrophoresis was first performed to identify the protein markers on EVs and the presence of exosomal protein markers TSG101, CD63, CD81, and ALIX was further confirmed by a semi-quantitative exosome-specific antibody array. Subsequently, we compared the cellular uptake behaviors of NP-loaded EVs with free NPs using MDA-MB 231br cell line by iron staining. The iron staining suggested that free NPs were readily taken up by cancer cells via endocytosis and localized at certain areas within the cells while uniform iron staining across cells was observed for cells treated with NP-loaded EVs.

## 2. Materials and Methods

### 2.1. Materials

All the chemical reagents were commercially purchased and used without further purification: ferric chloride anhydrous (ACROS, 98%), quercetin dihydrate (Alfa Aesar, 97%), Dulbecco’s phosphate buffer saline without calcium and magnesium (Corning), methanol (Alfa Aesar, 100%), Dulbecco’s Modified Eagle Medium (DMEM, ATCC), Dulbecco’s phosphate buffer saline 10× (Gibco), MES-SDS running buffer (Thermofisher Scientific, Waltham, MA, USA), LDS sample buffer (Thermofisher scientific), fetal bovine serum (FBS, ATCC), and penicillin/streptomycin (Thermofisher Scientific). Bradford Reagent (Thermofisher Scientific), Exo-Check exosome antibody array kit (System Biosciences, Palo Alto, CA, USA). Aqueous solutions were fabricated with nanopure water from Thermo Scientific Barnstead Nanopure water system. 

### 2.2. Preparation of Quercetin-Iron Complex Nanoparticles

The QFeNPs were synthesized by crosslinking Q-Fe complexes with excess iron ions. Briefly, 10 mM quercetin dihydrate stock solution (0.604 g of quercetin dihydrate in 200 mL of methanol) and 10 mM iron chloride stock solution (16 mg FeCl_3_ in 10 mL nanopure water) were first prepared. Then, the quercetin stock was adjusted to pH 9 with an equal volume of 500 mM HEPES. To facilitate NP formation, a molar ratio of quercetin to iron of 1:5 was used by mixing iron chloride stock solution (2.5 mL) with 0.25 mL pH-adjusted quercetin solutions. The reaction mixture was then heated at 60 °C for 2 h while stirring. Finally, the NPs were collected by centrifugation (15,000× *g* rpm, 15 min). To sterilize the NPs, 70% ethanol was added to the nanoparticles for 15 min. Then the NPs were dispersed in sterile PBS at 4 mg/mL.

### 2.3. Cell Culture

MDA-MB-231br, a brain metastatic variant of breast cancer cell line, MDA-MB-231, was obtained from Dr. Shreyas Rao’s laboratory at The University of Alabama. The cells were grown in Dulbecco’s Modified Eagle Medium with 10% fetal bovine serum and 1% penicillin/streptomycin. For NP treatment, 3 × 10^5^ cells were seeded in a T-25 flask with 3 mL of growth media and checked for confluency every 24 h. To prepare NP-loaded EVs, cells were treated with 200 µL of NPs (4 mg/mL) when cells reached 50% confluency. To prepare empty EVs, we cultured another flask seeded with same number of cells as a control and added same volume of 1 × PBS instead of NPs when cells were at 50% confluency. After 24 h incubation, the growth media was aspirated out, and replaced with 3 mL of serum-free growth media. After an additional 24 h, the growth media solution was collected for EV isolation.

### 2.4. EV Isolation with Direct Flow Filtration

After 24 h incubation, the serum-free growth medium (~3 mL) was collected into a 50 mL centrifuge tube and diluted with 1× PBS to 20 mL. The dilution was necessary to reach the minimal volume required for DFF. Prior to DFF, cell debris and cellular proteins were first removed by two cycles of centrifugation (1000× *g*, 10 min, and 10,000× *g*, 10 min) at 4 °C. The cleaned supernatant was transferred to a Sterlitech HP4750 DFF system with a membrane active area of 14.6 cm^2^ and filtered with a 100 kDa molecular weight Synder polyethersulfone (PES) membrane filter. The applied pressure was maintained at 0.97 bar, and the solution was stirred at ~400 rpm, which resulted in an average feed flow rate of ~0.75 mL/min based on three replications. Finally, the EVs were collected by soaking the membrane in 5 mL of sterile 1 × PBS and then filtered using a 200 nm Avanti extruder 5 times. The resulting EV PBS solution was further used for characterization and cellular uptake studies. NP-treated cells grown under healthy conditions were used as a comparison. For EV yield quantification, the membrane after DFF was soaked in nanopure water rather than PBS, and then freeze-dried. The average amounts of EVs from three experiments were used for yield quantification.

### 2.5. SDS-PAGE Gel Electrophoresis

To confirm the presence of EVs, SDS-PAGE gel electrophoresis was performed to evaluate typical EV protein markers using NuPAGE 4–12% bis-tris 1 mm mini protein gel. In brief, 90 µL of EVs isolated by DFF were mixed with 30 µL of an aqueous LDS sample buffer (10% glycerol, 2% LDS, 0.51 mM EDTA, 141 mM Tris-base, 106 mM Tris-HCl, 0.22 mM SERVA Blue G-250) and heated up to 70 °C for 5 min. The gel cassette was submerged in an aqueous MES SDS running buffer (50 mM MES, 50 mM Tris base, 0.1% SDS, 1 mM EDTA; pH 7.3). A total of 30 µL of the sample along with 10 µL protein markers were loaded into loading wells, respectively. After electrophoresis for 45 min at 120 V, the gel was incubated in a staining container containing an aqueous Coomassie blue staining solution (0.1% Coomassie blue, 40% ethanol, and 10% acetic acid). To facilitate gel staining, the gel in the staining solution was microwaved for 1 min at 1000 W and then shaken at room temperature for 1 h. After removing the staining solution, the gel was rinsed twice with nanopure water. Finally, the gel was submerged in an aqueous de-staining solution (10% ethanol and 7.5% acetic acid) to remove the stain.

### 2.6. ExoCheck Antibody Array

Characteristic exosomal protein markers were identified using a semi-quantitative exosome-specific antibody array system following the manufacturer’s instructions (System Biosciences Inc.). The antibody array kit comes with 12 pre-printed spots including 8 antibodies for known EV markers (CD63, CD81, ALIX, FLOT1, ICAM, EpCam, ANXA5, and TSG101) and 4 controls, including 2 positive controls (HRP detection), a blank spot (background control) and GM130 cis-Golgi marker, which monitors any cellular contamination. Briefly, 50 µg (100 µL, 0.5 mg/mL concentration) of EVs in 1× PBS was mixed with 10× lysis buffer from the kit and mixed with 1 µL of the labeling reagent. The mixture was incubated for 30 min at room temperature with constant mixing. Labeled EV lysates were then passed through packed bed columns to remove excess labeling agent, followed by blocking with 5 mL of blocking buffer. ExoCheck array membrane was first cleaned with nanopure water and then incubated with labeled EV lysate/blocking buffer mixture for 1 h at room temperature. After 1 h, the membrane was washed twice using 1× washing buffer and then incubated with detection agent for 30 min at room temperature. The membrane was washed twice again with washing buffer. Supersignal West Pico Luminol Enhancer Substrate was used to detect chemiluminescence. The blot was imaged using an iBright FL1500 imaging system. 

### 2.7. Cellular Uptake and Prussian Blue Staining

Cells were seeded in six-well plates at a density of 2 × 10^5^ cells with 2 mL of medium per well. The cells were incubated at 37 °C for 48 h to allow them to attach fully. Then, the growth media was replaced with new media containing 800 µg of QFeNPs or NP-loaded EVs (125 µg). Cells were collected at 6 h and 24 h to examine the cellular uptake by Prussian blue staining. Cells treated with PBS were used as a control. After washing twice with 1 mL of PBS, the collected cells were fixed with an aqueous 4% paraformaldehyde solution in PBS (1 mL) and incubated for 15 min at 37 °C. Then, the fixed cells were washed again with PBS (1 mL) once and then treated with the Prussian blue staining kit as per standard protocol. Briefly, cells were treated with an aqueous 4% HCl solution (0.6 mL), 3 times that of the NP amount fed to the cells (0.2 mL), and then incubated for 30 min. Then, 0.6 mL of potassium ferrocyanide was added and incubated for 30 min. After washing with 1 mL of PBS, the treated cells were incubated in 1:1 solution of PBS (0.5 mL) and nuclear fast red dye (0.5 mL) for 10 min. Finally, the cells were washed again with 1 mL of PBS and imaged using an inverse compound light microscope.

### 2.8. Characterization of Nanoparticles and EVs

The size and morphology of QFeNPs or NP-loaded EVs were studied by TEM (Hitachi 7860). The nanoparticle-loaded EVs can be directly visualized while the empty EVs were fixed and stained prior to imaging. The hydrodynamic size in PBS was measured using a Malvern (Malvern, UK) Zetasizer Nano series dynamic light scattering instrument. During the TEM analysis, the NP-loaded EVs were in a dehydrated state on TEM grids without additional staining while the DLS measured the hydrated state of the EVs in the solution. To obtain the TEM of empty EVs, EVs (10 μL) were first fixed with 4% paraformaldehyde (10 µL) aqueous solution (1:1) for 30 min at 4 °C. The fixed EVs were then dropped on carbon–formvar-coated copper grids (300 mesh) and dried for 20 min. Filter paper was used to wick away excess liquid. The grid was then washed in deionized water (10 µL) 3 times, 1 min each. Then the grid was placed in a drop (10 μL) of 2% uranyl acetate and was kept in the dark for 5 min. Finally, the grid was washed again with DI water 3 times, 2 min each, and then visualized under the transmission electron microscope. FTIR spectra were collected on an Agilent Cary 630 FTIR Spectrometer equipped with an attenuated total reflectance (ATR) crystal by accumulation of 4 scans, with a resolution of 2 cm^−1^. The Prussian Blue Stained cells were imaged using a Ts2R, Nikon inverse compound light microscope using NIS-Elements imaging software (Nikon Instruments Inc.).

## 3. Results

### 3.1. Quercetin Iron Complex Nanoparticles

Figure 1A shows a representative TEM image of QFeNPs with a particle size around 8 nm. However, the hydrodynamic size of these QFeNPs in water was much larger as indicated by the DLS plot with an average size of 85 nm and a wide size distribution (Figure 1B). Figure 1C presents the FTIR spectra of QFeNPs and free Q. Compared with the FTIR spectrum of free Q, a new peak was observed at 440 cm^−1^ from the FTIR spectra of QFeNPs. In addition, the FTIR spectrum of QFeNPs showed several evident IR band shifts or disappearance. Peaks at 1320, 1380, 1160, 720, and 600 cm^−1^ all nearly disappeared. Peak broadening was observed around 1600, 1560, 1010, and 817 cm^−1^. Appendix A shows the full FTIR spectra of the QFeNPs and free quercetin in the range of 4000–400 cm^−1^. 

### 3.2. EV Preparation and Isolation by Direct Flow Filtration

The entire EV preparation and isolation process is illustrated in Figure 2A. First, MDA-MB-231br cells were treated with QFeNPs. After 24 h NP treatment, cell growth media was replaced with serum-free growth media to facilitate EV generation. After additional 24 h incubation, the growth media containing secreted EVs were collected. To isolate EVs, cell debris and larger cellular proteins were first removed with two cycles of centrifugation (1000× *g*, 10 min followed by 10,000× *g*, 10 min). The supernatant was then fed to the DFF system at a flow rate of 0.75 mL/min at 0.97 bar using a 100 kDa PES membrane. Figure 2B shows a representative TEM image of QFeNP-loaded EVs isolated using DFF, where the NPs were visible and contained by EV membranes. The EVs on TEM image were in a dehydrated state with a rough size range of 50–100 nm, but the DLS plot suggested the hydrodynamic size of the EVs had an average mean distribution of around 250 nm. As a comparison, QFeNP-loaded EVs were isolated using UC method, where the EVs were larger in size (>500 nm) with a wider size variation (Appendix A). The DLS plot further showed the broader size distribution of EVs isolated by UC with an average size distribution around 350 nm. In addition, a shoulder peak around 100 nm could clearly be seen and the plot tailed to the larger size, ~900 nm (Appendix A). 

Figure 3A shows a representative TEM image of empty EVs stained with 2% uranyl acetate obtained after being isolated by DFF. The DLS plot of the empty EVs exhibited an average mean hydrodynamic size distribution around 270 nm (Figure 3B). In contrast, the empty EVs isolated by UC method were much larger (Appendix A). Unfortunately, a quality DLS plot could not be obtained due to the low EV concentration and polydispersity. Finally, the EV yields from DFF and UC were quantified based on the average freeze-dried EV weights from three experiment replicas. The EV yield from DFF was significantly higher than UC (Figure 3C), where DFF had an EV yield of 0.83 ± 0.10 mg/mL cell culture while the EV yield from UC isolation was 0.23 ± 0.06 mg/mL cell culture. To rule out the collection of apoptotic bodies, we compared the NP-treated cells grown under healthy condition and serum-free condition for 24 h (Appendix A). With growth media remaining in the cell growing flasks, not much difference in cell morphology was observed for cells under both conditions. However, after removing the growth media, the cells under the stressed condition shrank, but remained attached to the flask (Appendix A). The cells under the healthy condition remained unchanged. 

### 3.3. EV Protein Marker Identification

The characteristic protein markers for QFeNP-loaded EVs were identified by SDS-PAGE gel electrophoresis and were further confirmed by a semi-quantitative ExoCheck antibody array (Figure 3D,E). On the SDS-PAGE gel, several protein bands were observed in the range of 20–100 kDa, which could be assigned to CD9 (~23 kDa), CD81 (~28 kDa), CD63 and/or TSG101 (40–60 kDa), hsp70 (~70 kDa), and ALIX (~95 kDa). The semi-quantitative ExoCheck antibody array suggested the presence of several typical EV markers, including CD63, CD81, TSG101, intracellular adhesion molecules 1 (ICAM), ALIX, and ANXA5. Very faint bands were observed for the epithelial cell adhesion molecule (EpCAM) and Flotillin (FLOT-1). Although the tetraspanin protein markers CD63 and CD81 exhibit faint bands compared with ALIX and TSG101, it was enough to determine that characteristic protein markers were present in the EVs. Because of the much lower yield of EVs from UC-isolation, there were not sufficient protein markers to produce detectable signals.

### 3.4. Cellular Uptake Studies

Figure 4 shows the Prussian blue iron staining images of MDA-MB-231br cells treated with QFeNP-loaded EVs and free QFeNPs at different treatment times. During the study, cells treated with PBS were used as a control (Figure 4A,D). Prior to Prussian blue staining, cells were washed extensively to remove any QFeNP-loaded EVs and free QFeNPs that were not taken up by cells. After 6 h of incubation, blue staining was barely seen for cells treated with QFeNP-loaded EVs (Figure 4B). In contrast, evident blue staining was clearly seen from cells treated with free QFeNPs after 6 h (Figure 4C). After 24 h of incubation, clear blue staining was observed from cells treated with QFeNP-loaded EVs (Figure 4E). In addition, the blue staining was more uniformly distributed across cells, compared with cells treated with free NPs (Figure 4F). In comparison, the blue staining of the cells treated with free QFeNPs were in blue patches and QFeNPs were localized in certain compartments of the cells (Figure 4F). The control cells treated with PBS did not show any blue staining (Figure 4A,D). 

## 4. Discussion

The objective of this study was to evaluate the effectiveness of using DFF to isolate NP-loaded EVs from cell culture. Here, MDA-MB-231br, a brain metastatic variant of breast cancer cell line, was used as a model system. This cell line is the most studied triple-negative breast cancer cell line, which has a significant practical value. For example, MDA-MB-231 cells have been extensively used for cancer metastasis [42,43] and drug resistance studies [44,45]. During this process, QFeNPs were used as the loading NPs for several reasons. First, the QFeNPs can serve as a potential imaging agent for magnetic resonance imaging tracking [46]. Compared with inorganic iron oxide NPs, the QFeNPs are fabricated of molecular Q-Fe complex, which can be easily degraded and cleared. Second, the coordination ligand, Q, is known to have anticancer and anti-inflammatory activities, and the ability to reverse multi-drug resistance [47,48]. The multiple chelation sites of Q also allow for coordination with metal ions, resulting in increased bioavailability, biological activities, and stabilities of Q [46,49,50,51]. Solubility and stability are some of the major issues limiting biomedical applications of Q [52]. The NP formation also leads to enhanced cellular uptake and effective delivery [53]. The smaller size selection (<15 nm) of QFeNPs were mainly for easy clearance for potential in vivo applications based on previous work of iron oxide NPs [54]. 

### 4.1. Quercetin Iron Complex Nanoparticles

The QFeNPs were synthesized by simply mixing a Q-methanol solution (pH = 9) with an iron chloride (FeCl_3_) solution at a quercetin-to-Fe^3+^ molar ratio of 1:5 following our previously published protocols [55]. Here, methanol was used as the solvent for Q because it is insoluble in water. The adjusted pH of 9 facilitated the deprotonation of the hydroxyl groups of Q and subsequent coordination with iron ions. The excess iron ions also facilitated crosslinking of Q molecules, leading to NP formation as shown in Figure 1A. Compared with the 8 nm TEM image size, the increased hydrodynamic size of QFeNPs in water from the DLS plot (Figure 1B) was likely due to the hydrogen bond formation and possible π–π stacking of Q molecules on QFeNP surfaces. Even though the observed size in water appeared larger than that measured using TEM, the NPs in water maintained a uniform stable dispersion, suggesting that the NP formation increased the water solubility of Q. 

The comparison of FTIR spectra of QFeNPs and free Q provided information on the coordination between Q and iron (Figure 1C). Close attention was provided to the most known coordination sites of catechol groups, hydroxyl at carbon 3 and 5 (C5/C3) positions, and the neighboring ketocarbonyl group. The newly appeared peak at 440 cm^−1^ was assigned as the characteristic Fe-O stretching, suggesting Q-Fe complex formation. Further, the evident IR band shifts or disappearance indicated that the iron coordination might involve -C-OH and C=O. For instance, for C=C-OH groups, the hydroxyl C-O stretching at 1320 and 1380 cm^−1^, hydroxyl C-OH in-plane (600 cm^−1^) and out-of-plane (720 cm^−1^) bending, and hydroxyl CO-H bending (1160 cm^−1^) all nearly disappeared. For the -C=O group, the peaks of -C=O stretching at 1600 and 1560 cm^−1^, and -C=O in-plane (1010 cm^−1^) and out-of-plane (817 cm^−1^) bending became broadened, suggesting its involvement in binding. However, the specific binding sites were difficult to differentiate from the FTIR spectra. These well-characterized NPs were sterilized in an aqueous 70% alcohol and re-dispersed in PBS for subsequent cellular uptake studies.

### 4.2. EVs Preparation and Isolation by Direct Flow Filtration

Our previous NP uptake studies using different types of NPs and various cell lines suggested that NPs were effectively taken up by cells after 24 h incubation, mainly through endocytosis [56,57,58]. Therefore, 24 h NP treatment was selected to produce QFeNP-loaded EVs. After 24 h NP treatment, cell growth media was switched to serum-free growth media, a known stressed growth condition to facilitate EV formation [29,58,59]. Cell-secreted EVs are mainly present in the growth media. Therefore, EVs in growth media rather than cell lysates were mainly isolated using DFF and the traditional UC isolation was used as a comparison. To prevent the formation of apoptotic bodies, NP-treated cells were maintained in serum-free media for 24 h incubation when most of the cells were healthy. The two centrifugation cycles (1000× *g*, 10 min followed by 10,000× *g*, 10 min) sufficiently removed the majority of cellular protein particles, as indicated by the Bradford assay before and after centrifugation (Appendix A). The low speed centrifugation prior to DFF also helped avoid potential protein contaminations and membrane fouling [60]. The EVs were then isolated by PES membrane with a median molecular weight cutoff (MWCO) of 100 kDa.

During the DFF process, the applied pressure pushes the fluid through the membrane along with substances smaller than the membrane pore size while larger particles remain on the membrane surface [33]. The selection of PES membrane was for its demonstrated very low protein binding capacity in flow filtration [61,62]. Membrane filters with a MWCO of 100 kDa led to EVs of high uniformity, because EVs smaller than 40 nm are non-exosomal impurities [59]. Our major goal was high recovery of uniform EVs; therefore, a membrane with a MWCO substantially lower than that of the molecular weight of the EVs was used to ensure a high recovery of the targets. This selection was based on the general rule that a membrane with an MWCO that is 3 to 6 times lower than the molecular weight of the molecules to be retained provide high retention of the molecules [63]. Previous studies on ultrafiltration suggested that membranes with a pore size of 100 kDa MWCO showed the most efficient recovery [64]. In addition to the pressure and membrane pore size, the flow rate is another parameter that can affect the EV recovery. Generally, the lower the flow rate, the higher the recovery. Therefore, a flow rate of 0.75 mL/min was used. The EVs on the filtration membrane were collected by rinsing PES membrane with 1× PBS.

The TEM image of QFeNP-loaded EVs isolated using DFF in Figure 2B clearly showed the NPs contained within EV membranes. Compared with the EVs size range of 50–100 nm on the TEM image in a dehydrated state, the EV hydrodynamic size from DLS (around 250 nm) was much larger. The increased hydrodynamic size was likely due to hydration of EVs and the presence of surface proteins on the EVs. The difference between the TEM size and DLS size was normally observed for typical cell-derived EVs [65]. The presence of the QFeNPs in groups also indicated the integrity of the EVs, because free NPs would have passed through the DFF membrane. Empty EVs sometimes exhibited cup-shaped morphology on TEM image from EV collapse during drying from previous study [66]. However, the cup-shaped morphology was not observed for QFeNP-loaded EVs. The QFeNP-loading likely helped maintain the shape of EVs, which was consistent with other types of NP-loaded EVs [23].

Empty EVs isolated by the DFF method showed similar morphology and size to QFeNP-loaded EVs. In contrast, the empty EVs isolated by the UC method was much larger (Appendix A). The EV yield quantification showed 3.6 times more EVs isolated by DFF compared with UC isolation. The much-increased yield was a significant benefit of DFF isolation compared with the UC method in EV isolation. The comparison of the NP-treated cells grown under healthy condition and serum-free condition for 24 h also helped rule out the collection of apoptotic bodies, cells under both conditions were alive after 24 h treatment. However, a morphological change was observed for NP-treated cells grown in a stressed condition. 

### 4.3. EV Protein Marker Identification

Several protein markers are normally enriched in the transmembrane surface of EVs, such as ALIX, TSG101, hsp70, CD63, CD9, and CD81. These proteins serve as characteristic markers to characterize EVs derived from cell culture [32,59,67]. For example, protein markers, ALIX, hsp70, CD63, TSG101, CD9, and CD81 were observed in MDA-MB-231 cell-derived EVs [68,69]. The reported molecular weights of the protein markers varied depending on the cell types, processing conditions, etc. [70,71]. For example, the CD63 protein marker showed differences in MCF7 and MDA-MB-231 cells, although both of them are breast cancer cell lines [63]. In another study, this membrane protein showed two distinct bands in the range of 30–60 kDa for EVs from MDA-MB-231 breast cancer cells [72], but 54 kDa is the most reported molecular weight. In addition to CD63, ALIX and TSG101 are commonly used as protein markers for EVs [73], and for verification of EV isolation [74]. The characteristic protein markers for QFeNP-loaded EVs were studied by SDS-PAGE gel electrophoresis and further confirmed by a semi-quantitative ExoCheck antibody array (Figure 3D,E). According to previous studies, the protein bands observed in a broad range of 20–100 kDa on the stained SDS-PAGE gel can be assigned to CD9 (~23 kDa), CD81 (~28 kDa), CD63 and/or TSG101 (40–60 kDa), hsp70 (~70 kDa), and ALIX (~95 kDa). Several typical markers were confirmed in the specific antibody array, including CD63, CD81, TSG101, intracellular adhesion molecules 1 (ICAM), ALIX, and ANXA5. Very faint bands were observed for epithelial cell adhesion molecule (EpCAM) and Flotillin (FLOT-1). Although the tetraspanin protein markers CD63 and CD81 exhibit faint bands compared with ALIX and TSG101, it was sufficient to determine that characteristic protein markers were present in the EVs. The detection of all these markers indicated that our QFeNP-loaded EVs exhibited at least one protein from each category of transmembrane, cytosolic and intracellular proteins, which is a characteristic feature of EV isolation as found in previous studies [75,76].

### 4.4. Cellular Uptake Studies

Previous studies by others suggested that cell-derived EVs could selectively target their cells of origin [77]. Therefore, cell uptake behaviors of QFeNP-loaded EVs were studied using MDA-MB-231br cells and compared with free QFeNPs to evaluate their targeting capabilities. Because of the presence of iron within the QFeNPs, Prussian blue iron staining offered a facile way to monitor cellular uptake. Our previous NP uptake studies using different NPs and various cell lines have suggested that evident NP uptake can be observed after 4 h and NP uptake reaches the maximum around 24 h [56,57,58]. Therefore, we selected 6 h and 24 h incubation periods for our cellular uptake studies. Prior to Prussian blue staining, cells were washed extensively to remove any QFeNPs or NP-loaded EVs that were not taken up by cells. After 6 h incubation, free QFeNPs exhibited evident cellular uptake, but not the QFeNP-loaded EVs. This observation is likely due to the much lower QFeNP concentration in the QFeNP-loaded EVs compared with the free QFeNPs. After 24 h incubation, both free QFeNPs and QFeNP-loaded EVs showed sufficient cellular uptake. However, the blue staining was more uniformly distributed across cells treated with QFeNP-loaded EVs while the blue staining of cells treated with free QFeNPs was in blue patches and NPs were localized in certain compartments of the cells. Previous studies have shown that cargos in EVs can be delivered via endocytosis or fusion into cytosol [7,78]. We also observed evident cell death from free QFeNP-treated cells, inferring the anticancer activities of these NPs as shown in previous reports [47]. NPs taken by cancer cells via endocytosis have been observed by many studies [56,57,58], which have localized NPs inside endosomes. Therefore, we believe that QFeNP-loaded EVs likely contained mainly exosome subpopulation with minimal apoptotic bodies from dead cells. This is also consistent with the biomarker analysis with clear TSG101, CD81 and ALIX band but faint FLOT-1 band.

## 5. Conclusions

In summary, we successfully prepared QFeNP-loaded EVs via DFF method. Compared with UC isolation, a significant increase in EV yields was observed for DFF (~4 times increase). SDS-PAGE gel electrophoresis of QFeNP-loaded EVs exhibited several protein bands in the region of typical characteristic EV protein markers. The presence of specific EV protein markers, such as ALIX, TSG101, CD63, and CD81, was confirmed using ExoCheck antibody array. Cellular uptake analysis using Prussian blue iron staining suggested that the isolated EVs could be effectively taken up by cells of origin. Our studies demonstrated the feasibility of using DFF for EV isolation with high recovery rate. The therapeutic application of EVs as drug nanocarriers potentially advances metastatic cancer treatment as EVs have the ability to cross several biological barriers. Future studies will investigate the in vivo tracking of EVs because of the MRI tracking potential of QFeNPs-loaded EVs.

## Figures and Tables

**Figure 1 pharmaceutics-15-01551-f001:**
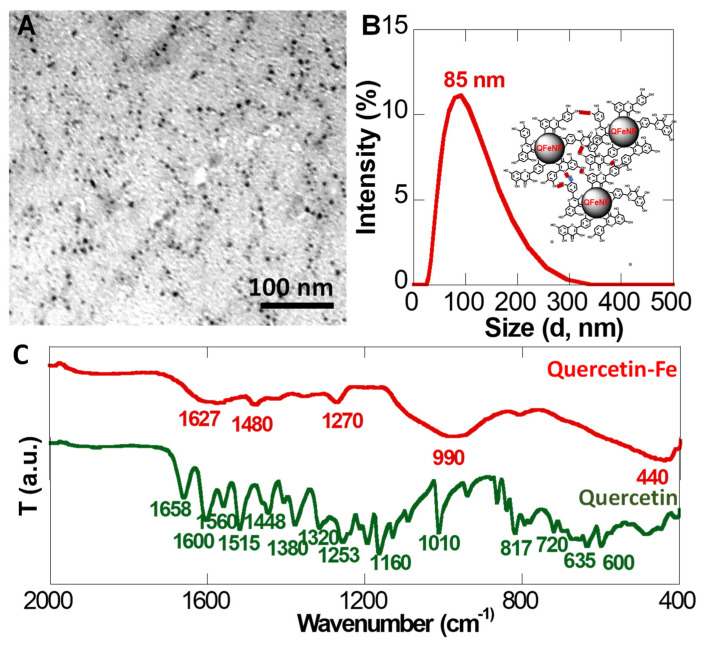
Quercetin-iron complex nanoparticles. (**A**) a representative TEM image, (**B**) a DLS plot and illustration of NP interactions via H-bonds and possible π–π stacking, and (**C**) ATR-FTIR spectra of quercetin and quercetin-iron complex nanoparticles.

**Figure 2 pharmaceutics-15-01551-f002:**
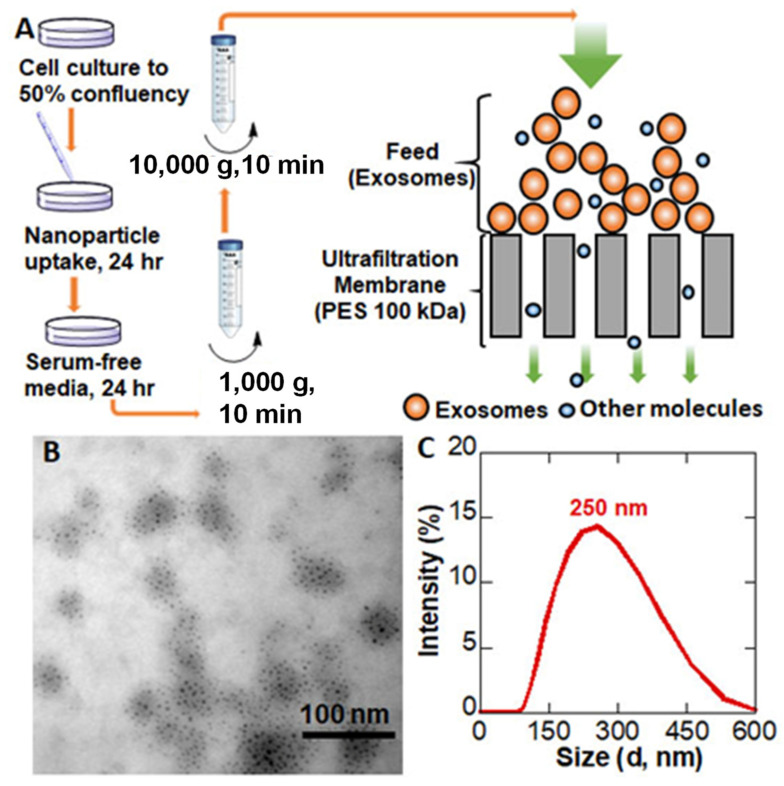
Nanoparticle-loaded EVs: (**A**) an illustration of EV preparation process, (**B**) a representative TEM image of EVs isolated by DFF, (**C**) a DLS plot.

**Figure 3 pharmaceutics-15-01551-f003:**
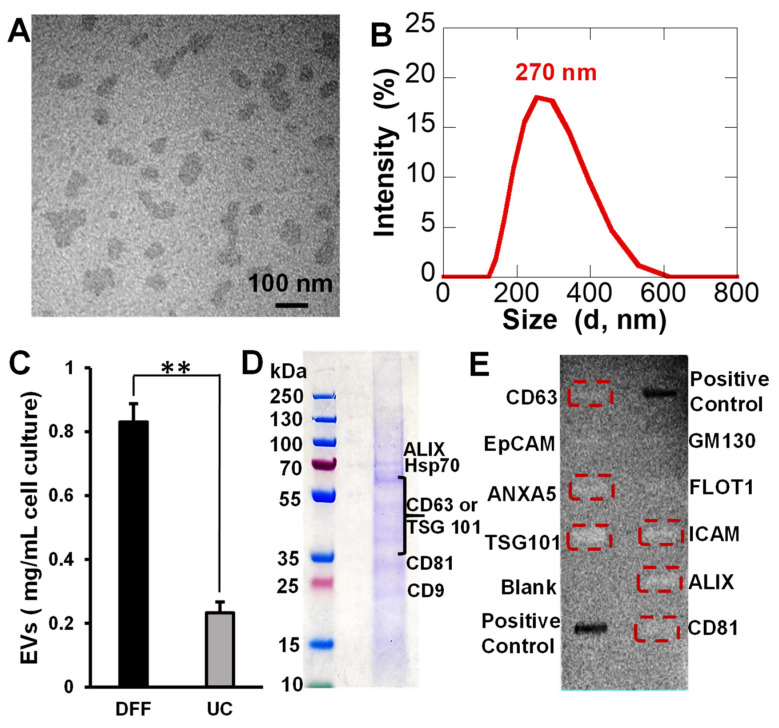
(**A**) A TEM image of empty EVs from DFF isolation without NP treatment. (**B**) DLS plot of empty EVs, (**C**) comparison of EV yields from DFF and UC, (**D**) SDS-PAGE gel image of NP-loaded EVs with several protein bands, and (**E**) EV protein marker analysis of NP-loaded EVs using a semi-quantitative ExoCheck antibody array containing 8 types of specific antibodies and controls. The average EV weight per mL of cell culture collected via DFF and UC were plotted with SEM. Data were calculated using t-test with Welch’s correction in GraphPad Prism 9.4.1. *n* = 3, ** *p* < 0.01.

**Figure 4 pharmaceutics-15-01551-f004:**
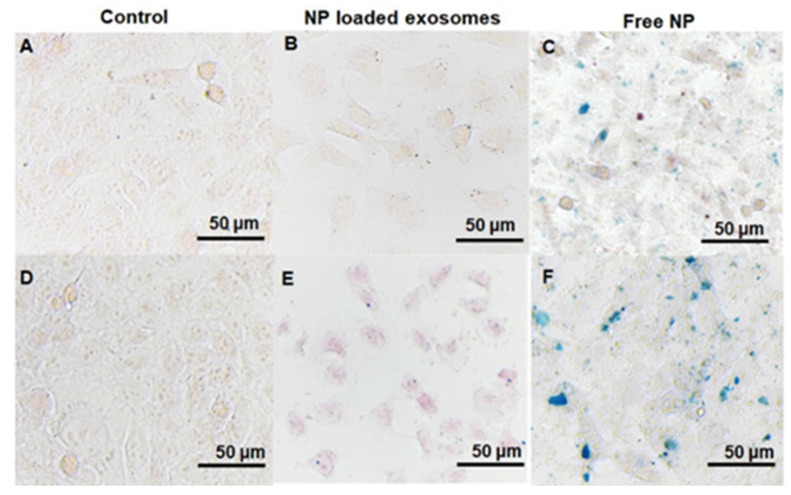
Prussian blue staining of cells treated with PBS, NP-loaded EVs, and free QFeNPs: (**A**) 6 h with PBS, (**B**) 6 h with NP-loaded EVs, (**C**) 6 h with free QFeNPs, (**D**) 24 h with PBS, (**E**) 24 h with NP-loaded EVs, (**F**) 24 h with free QFeNPs.

## Data Availability

All critical data are reported either in the article or Appendix A. The data reported here are also available upon request from the corresponding author.

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
