# Peer review of "Preparation of Nanoparticle-Loaded Extracellular Vesicles Using Direct Flow Filtration"

_pharmaceutics, 2023, doi:10.3390/pharmaceutics15051551_

Round 1

Reviewer 1 Report

Drug loading and precise isolation of exosomes are among the major challenges in exosome-based therapeutics. In this manuscript, the authors present a new approach for efficiently loading quercetin-Fe complex nanoparticles into exosomes secreted from breast cancer cells (MDA-MB231br). However, the data presented in the manuscript does not strongly support the conclusions, and there are inconsistencies with the widely accepted principles of exosomes, particularly in terms of size, morphology, and isolation methods. To improve the manuscript's quality, we recommend the following:

In Figure 1A, it is important to show scheme that can explain the disparities in size of the nanoparticles shown in the TEM image and DLS measurement. The authors should also provide a rationale for selecting this particular size of nanoparticles.

In Figure 2B, the authors should compare the TEM images of EVs isolated by ultracentrifugation to those isolated by the DFF method. They should also explain why most EVs lost their cup-shaped structures, which have been reported in other manuscripts using similar methods to visualize EV morphology. Furthermore, the authors should provide the DLS size of EVs from ultracentrifugation and the encapsulation rate of nanoparticles in EVs isolated by ultracentrifugation, and compare them to those obtained by the DFF method.

To accurately describe the particles isolated by the DFF method, it is more appropriate to use the term "extracellular vesicles" instead of "exosomes".

In Figure 3D, the authors should clearly label the lanes, and in Figure 3E, they should highlight the distinct bands, which are difficult to see. The authors should also clearly label which band represents the control obtained by the ultracentrifugation method. Figure 3E is difficult to comprehend, and in Figure 3C, it would be more informative to know the milligrams of exosomes isolated from each milliliter of medium rather than from each batch.

The manuscript has a good English language quality.

Reviewer 2 Report

The authors presented very interesting work with nicely described methods. In general NPs preparation and characterization were presented in the previous article: https://doi.org/10.3390/pharmaceutics15041041 whereas this work is an extension of it. Exosome preparation as a carrier system is a very interesting topic with still lots of unknowns.

I just have one question that bothers me for some time. The development of Iron-based nanoparticles seems to be exploited by many researchers and then pointed to as a way of drug delivery. As far as I know, iron ions themselves might produce lots of adverse or toxic events on cells so making carriers from them might be not suitable for drug development. This is just my opinion but can you elaborate on it in the discussion as far as you point results of your work as potential DDS.

Can you put numerical results for the yield of the process in the text? I know there is a plot but numerical values with sd are better for further comparisons.

Round 2

Reviewer 1 Report

The authors have successfully addressed all the concerns and incorporated the major recommendations. Thank you for the substantial effort put into this revision. The manuscript is now suitable for publication.